# The C-Terminus of the PSMA3 Proteasome Subunit Preferentially Traps Intrinsically Disordered Proteins for Degradation

**DOI:** 10.3390/cells11203231

**Published:** 2022-10-14

**Authors:** Assaf Biran, Nadav Myers, Shirel Steinberger, Julia Adler, Marianna Riutin, Karin Broennimann, Nina Reuven, Yosef Shaul

**Affiliations:** Department of Molecular Genetics, Weizmann Institute of Science, Rehovot 76100, Israel

**Keywords:** intrinsically disordered proteins, proteasomal degradation, 20S proteasome, proteostasis

## Abstract

The degradation of intrinsically disordered proteins (IDPs) by a non-26S proteasome process does not require proteasomal targeting by polyubiquitin. However, whether and how IDPs are recognized by the non-26S proteasome, including the 20S complex, remains unknown. Analyses of protein interactome datasets revealed that the 20S proteasome subunit, PSMA3, preferentially interacts with many IDPs. In vivo and cell-free experiments revealed that the C-terminus of PSMA3, a 69-amino-acids-long fragment, is an IDP trapper. A recombinant trapper is sufficient to interact with many IDPs, and blocks IDP degradation in vitro by the 20S proteasome, possibly by competing with the native trapper. In addition, over a third of the PSMA3 trapper-binding proteins have previously been identified as 20S proteasome substrates and, based on published datasets, many of the trapper-binding proteins are associated with the intracellular proteasomes. The PSMA3-trapped IDPs that are proteasome substrates have the unique features previously recognized as characteristic 20S proteasome substrates in vitro. We propose a model whereby the PSMA3 C-terminal region traps a subset of IDPs to facilitate their proteasomal degradation.

## 1. Introduction

Proteasomal protein degradation plays key roles in diverse cellular processes and cell fate determination [1,2]. The 26S proteasome is a major molecule which catalyzes protein degradation. Its 20S barrel-shaped proteolytic core particle is capped at one or both ends by the 19S regulatory complex [2,3,4,5]. The 20S proteasome is composed of four stacked rings, two PSMA and two PSMB rings, in a barrel shape. Proteolytic activity occurs in the chamber formed by the inner PSMB rings. The outer PSMA rings are identical. and each has seven distinct subunits. The N-termini of the PSMA subunits form a gated opening controlling substrate entry into the proteasome [6,7]. 

Proteins destined for degradation are first identified as “legitimate” substrates by the proteasomes [8,9]. This is accomplished by targeting the proteasome via protein–protein interaction. Polyubiquitin chains are covalently attached to the substrates, marking them target proteasomes for degradation [10]. The polyubiquitin chain binds the 19S regulatory particle of the 26S proteasome either directly or through proteins, which transiently associate with 19S [11,12,13,14]. To date, three of the subunits of the 19S particle, namely Rpn1, Rpn10 and Rpn13, have been identified as ubiquitin receptors [15,16].

The 20S proteasome regulatory particles, including 19S, PA28 and PA200, each interact with the PSMA ring, modulating proteasomal activity by opening the narrow entrance into the orifice and improving the accessibility of substrates into the catalytic chamber [17]. However, a number of studies suggest that 20S proteasome is functional in the absence of any regulatory particles, in a ubiquitin-independent manner, as has been demonstrated in vitro using highly purified 20S proteasome and substrates, and in the cells via the analysis of the proteasome-generated peptides [18,19,20,21,22,23]. The question of how proteins enter the 20S chamber that is predicted to be closed, remains unanswered. However, the intrinsically disordered proteins (IDPs) or proteins with disordered regions (IDRs) are preferential 20S substrates, also because these proteins are naturally unfolded [24,25]. Indeed, IDPs and oxidized proteins have been shown to undergo ubiquitin-independent proteasomal degradation, including by the 20S proteasomes [18,19,20,21]. Key questions are whether and how the proteasome recognizes these substrates. This study has found that PSMA3′s C-terminus binds multiple IDPs, and most of them are 20S substrates. Thus, we propose a model whereby the 20S catalytic particle has a substrate receptor to trap certain IDPs for proteasomal degradation. 

## 2. Materials and Methods

### 2.1. Cell Culture

The human embryonic kidney cell line HEK293, the human osteosarcoma cell line U2OS and the colon cancer cell line HCT116 were grown in DMEM supplemented with 8% fetal bovine serum (both from GIBCO, Life Technologies, Thermo Scientific, Waltham, MA, USA), 100 units/mL penicillin and 100 μg/mL streptomycin (both from Biological Industries, Beit-Haemek, Israel) and cultured at 37 °C in a humidified incubator with 5.6% CO_2_. 

### 2.2. Plasmids and Transfection 

PSMA subunits (kindly provided by Prof. K. Tanaka, Tokyo Metropolitan Institute of Medical Science, Japan) were cloned into pBiFC-VN173 (Addgene plasmid no. 22010), a gift from Prof. Chang-Deng Hu (Purdue University). The 6xmyc p21 was cloned into pBiFC-CC155 (Addgene plasmid no. 22015), a gift from Prof. Chang-Deng Hu. Luc1 and Luc2, the respective N’ terminal and C’ terminal fragments of split Gaussia luciferase, were kindly provided by Prof. Adi Kimchi (Weizmann Institute of Science, Israel.). HEK293 cells were transfected using the calcium phosphate method [26]. U2OS cells stably expressing the chimeric PSMA3 subunit were created using the Gateway cloning system (Invitrogen, Carlsbad, CA, USA). 

### 2.3. Immunoblot Analysis

Cells were lysed with NP40 buffer (20 mM Tris-HCl pH7.5, 320 mM sucrose, 5 mM MgCl_2_, 1% NP40), supplemented with 1 mM dithiothreitol (DTT) and protease and phosphatase inhibitors (Sigma, St. Louis, MO, USA). A Laemmli sample buffer (final concentration 2% SDS, 10% glycerol, 5% 2-mercaptoethanol, 0.01% bromophenol blue and 0.0625 M Tris-HCl pH6.8) was added to the samples, heated at 95 °C for three minutes and loaded on a polyacrylamide–SDS gel [27]. Proteins were transferred to 0.45 μm cellulose nitrate membranes. Antibodies: mouse anti-tubulin and mouse anti-human p53 Pab1801 were purchased from Sigma (St. Louis, MO, USA). Mouse anti-HA antibody was purchased from Covance, Berkeley, CA, USA. Mouse anti-myc was produced by the Weizmann Institute Antibody Unit. Rabbit anti-PSMD1, a subunit of the 19S proteasome, was purchased from Acris (Herford, Germany). Rabbit anti-PSMA4, a subunit of the 20S proteasome [28], was kindly provided by Prof. C. Kahana, Weizmann Institute of Science, Israel. Secondary antibodies used were horseradish peroxidase-linked goat, anti-rabbit and anti-mouse (Jackson ImmunoResearch, West Grove, PA, USA). Signals were detected using the EZ-ECL kit (Biological Industries, Beit-Haemek, Israel).

### 2.4. Co-Immunoprecipitation 

Cell lysates were incubated with a primary antibody, as detailed in the relevant panels and figure legends, for 16 h. Samples were washed six times with NP40 buffer. Bound and associated proteins were eluted with Laemmli sample buffer or HA peptide (Sigma, St. Louis, MO, USA) according to the standard protocol. 

### 2.5. Nondenaturing PAGE

Samples were prepared and run as described [29,30].

### 2.6. Protein-Fragment Complementation Assays (PCAs) 

We employed the protein-fragment complementation approach [31]. To directly visualize p21 interaction with PSMA3 in the cells, we adopted the bimolecular fluorescence complementation (BiFC) assay [32]. In this assay, the GFP-reporter fluorescent protein (FP) is split into two fragments; the C-terminus FPC and the N-terminus FPN, which emit a fluorescent signal upon their interaction. Cells were co-transfected with PSMA3 subunit-FPC, potential substrate-FPN and H2B-red fluorescent protein (RFP), the latter used to identify the transfected cells. Cells with successful BiFC are green (VFP), while H2B-RFP makes the cell nuclei red. For flow cytometry analysis, cells were harvested 48 h post-transfection, washed and resuspended in PBS. Samples were analyzed with a BD LSR II flow cytometer using FACSDiva software v9.0 (BD Biosciences). The fluorescent intensities of VFP and RFP in RFP-positive cells were recorded and extracted using FlowJo software (FlowJo, LLC). The BiFC signal was normalized to RFP signal per cell. The BiFC/RFP median was used because the ratio distribution is skewed [33].

We also used split Gaussia luciferase. In this system, the receptor protein was fused to the C’-terminal fragment of Gaussia luciferase (denoted Luc2) and p21 fused to the N’- terminal fragment of the luciferase (denoted Luc1). Upon interaction, the luciferase enzyme reconstructs and its activity, which generates bioluminescence in the presence of its substrate, can be detected. Cell lysates were plated in a 96-well white plate, with 30 μL per well. Using automated injection in a Veritas microplate luminometer (Promega, Madison, WI, USA), the lysates were mixed with luciferase substrate. The solution was prepared by diluting coelenterazine (Nanolight Technology, Pinetop, AZ, USA) to a final concentration of 20 μM in an assay buffer (25 mM Tris pH 7.75, 1 mM EDTA, 0.6 mM reduced glutathione, 0.4 mM oxidized glutathione and 75 mM urea). A bioluminescent signal was read after injection of 100 μL of substrate solution and integrated over 10 s. 

### 2.7. GST Pulldown and MS Analysis

HEK293 extracts were prepared and heat- (95 °C for 5 min) treated as previously described [18]. The extracts were MS analyzed and were devoid of proteasome subunits. Next, we conducted GST pulldown assay [34]. To this end, we fused residues 187–255 and residues 188–241 of PSMA3 and PSMA5 C terminus, respectively, to GST to generate recombinant Trapper–GST fusion proteins. Recombinant GST proteins bound to glutathione agarose were incubated in a rotator with treated or naïve cell lysate for 16 h at 4 °C. Beads were washed six times with 300 μL NP40 buffer and recombinant GST and associated proteins were eluted from glutathione agarose beads with 70 μL of 10 mM glutathione in 50 mM Tris-HCl pH 9.5. 

The heat-treated HEK293 extracts and the GST–PSMA3-trapped proteins were MS analyzed after tryptic digestion, as described [18]. The mass spectrometry proteomics data have been deposited to the ProteomeXchange Consortium via the PRIDE partner repository with the dataset identifier PXD024791 and 10.6019/PXD024791.

### 2.8. In Vitro Degradation Assay

Heat-treated (95 °C for 5 min) cell extracts [18] were incubated with purified 20S proteasomes. The protocol of 20S purification is as previously described [18]. The proteasomes were not activated by SDS. Briefly, proteasomes were pelleted by centrifugation at 150,000 g, re-suspended and fractionated by a Hiprep Q column (anion exchange chromatography (Cytiva, Marlborough, MA, USA). The positive fractions were supplemented to 1.75 M (NH_4_)_2_SO_4_ and the remaining proteins were removed by pelleting at 15,000 g for 30 min. The supernatant was loaded on a HiTrap Phenyl HP column (high-resolution hydrophobic interaction chromatography (HIC)) (Cytiva, Marlborough, MA, USA). 

### 2.9. Data Analysis

Data analysis was performed with Rstudio version 1.1.456, R 4.1.3. and Microsoft Excel. FunRich software (http://www.funrich.org. Accessed date 8 February 2018) was used to plot proportional Venn diagrams. Gene ontology, prediction of structural disorder, PrLD, LCR and GR/PR di-peptide repeats interactor proteins were performed as previously reported [18]. The expected intersection between two groups was evaluated by a Z-–test with the null distribution calculated by a 10,000 simulation randomly choosing x from set A, y from set B and noting the intersection between the randomly chosen element. Protein sequences were obtained from UniProt (http://www.uniprot.org/ Accessed date 4 December 2018) using the annotated Swiss-Prot bank. Statistical tests and plotting were performed using MATLAB 2016b, The MathWorks, Natick, 2014.

## 3. Results

### 3.1. The 20S Proteasome PSMA3 Subunit Preferentially Binds IDPs 

We assumed that a subunit of the 20S complex could potentially act as an IDP trapper for degradation. To examine this possibility, we drew on interactome datasets, assuming the PSMA-interacting proteins were potential 20S substrates. Analyzing the IMEx data resource, which searches different databases of large-scale protein–protein interaction screens [35], PSMA3 and PSMA1 were found to be the preferred protein-binding constituents (Figure 1A,B). Next, using the IUPred algorithm [36], we evaluated the percentage disorder of the PSMA-interacting proteins, finding that the PSMA3-interacting proteins are uniquely highly enriched for IDPs (Figure 1C). We also compared PSMA3-interacting proteins found in the IMEx data resource to PSMA3-interacting proteins found in the HI.II.14 dataset from the human interactome project [37]. The interacting proteins found in the HI.II.14 dataset are also enriched with IDPs (Figure 1D). These analyses indicate that PSMA3 preferentially interacts with IDPs. 

### 3.2. PSMA3 Interacts with p21 in the Cells

The protein p21 is an IDP [39,40] that undergoes both ubiquitin-dependent and independent proteasomal degradation [41,42]. It has already been reported that p21 binds to the 20S PSMA3 subunit in vitro [43]. To directly visualize p21′s interaction with PSMA3 in the cells, we used the bimolecular fluorescence complementation (BiFC) assay. In this assay, the reporter fluorescent protein (FP) is divided into two fragments: the C-terminus FPC and the N-terminus FPN, which emit fluorescent signals upon their interaction. PSMA3 and PSMA5 subunits were fused to the FPC and co-transfected with a chimeric 6xMyc-p21 FPN. The Myc tag minimizes p21 proteasomal degradation [42], but should not affect PSMA3 association. The level of interaction between the 6xmyc p21 FPN and PSMA3–FPC was monitored by quantifying VFP-positive cells. 

First, we investigated whether the PSMA3–FPC chimera incorporates into the proteasomes using native gel analysis. According to the pattern of migration in the native gel, a fraction of the PSMA3–FPC successfully incorporated into 20S and 26S proteasome complexes (Figure 2A). To quantify the fraction of the incorporated chimeric PSMA3, we conducted successive proteasome depletion experiments (Figure 2B,C). The proteasomes were depleted from the cellular extract through the immunoprecipitation of the endogenous 20S proteasome PSMA1 subunit and were monitored for the presence of the chimeric PSMA3–FPC. We found that the PSMA3–FPC chimera was depleted as efficiently as the endogenous proteasome subunit PSMA1 (Figure 2D). As expected, under the same conditions, the 19S proteasome subunit PSMD1 was also depleted, although with lower efficiency. These results suggest that the vast majority of the PSMA3–FPC chimera protein is incorporated into the proteasomes. In the test of fluorescent signals, p21 FPN gave a stronger signal when co-transfected with PSMA3–FPC than did PSMA5–FPC (Figure 2E,F). Additionally, as a control, the structured NQO1 protein emitted a very poor signal. Constructs were expressed to the same level (Figure 2G). These data suggest that PSMA3 (but not PSMA5) interacts with p21 in the cells.

#### 3.2.1. The PSMA3 C-Terminus Is Sufficient to Interact with p21 

The PSMA subunits differ primarily at their C-termini [44]. We generated and examined PSMA3-truncated C-terminus mutants and found that a sharp reduction in PSMA3 and p21 interaction was obtained by the truncation of the C-terminal region (residues 187–255) (Appendix A). Since the PSMA3 C-terminus (Ct 187–255) is exposed to its surroundings in the context of the 20S and 26S proteasomes, it is accessible to the putative IDP substrates (Appendix A). Consequently, we assumed that PSMA3-Ct is the most likely p21-interacting region. To examine this possibility and to show that the PSMA3 C-terminus is sufficient to interact with p21, we constructed the PSMA5–PSMA3 chimeric constructs in which the C-terminus of PSMA5 was replaced with the homologous region of PSMA3. Two chimeric PSMA5–PSMA3 were constructed: one with a long PSMA3 Ct fragment (Ct 187–255) and the other with a shorter Ct187–229 fragment (Appendix A). The chimeric PSMA5–PSMA3 constructs had higher levels of GFP than the naive PSMA5 (Figure 3A), suggesting that the PSMA3 Ct is active in interacting with p21 even in the context of a truncated PSMA5. To validate the GFP data, we conducted co-immunoprecipitation experiments, for which we used a 6xmyc-tagged p21 construct lacking the FPN moiety. We found that p21 is co-immunoprecipitated with the chimeric PSMA5–PSMA3 constructs but not with the naive PSMA5 (Figure 3B). The reciprocal constructs in which the C-terminus of the PSMA3 was replaced by that of PSMA5 were constructed, but these expressed very poorly (Appendix A). These data suggest that the PSMA3-Ct is sufficient to interact with p21.

Next, we utilized the split luciferase reporter to monitor p21′s interaction with PSMA3 and with the PSMA3 Ct 187–255 fragment. The utilized constructs were efficiently expressed (Figure 3C). Luciferase activity was restored not only in the presence of full-length PSMA3, but also with the Ct 187–255 fragment (Figure 3D). These data suggest that the PSMA3 Ct 187–255 fragment binds p21 in vivo. Therefore, this region is a potential IDP trapper.

#### 3.2.2. The PSMA3 187–255 Fragment Interacts with Many IDPs 

To explore whether the p21 trapper is active in binding other IDPs, we generated chimeric GST–PSMA3 187–255 recombinant proteins. The chimera GST–PSMA5 188-241 and recombinant GST were used as controls (Figure 4A). We examined the recombinant trapper’s ability to pull down specific endogenous IDPs, such as c-Fos and p53, and ectopically expressed 6xmyc p21. We found that PSMA3 187–255 specifically pulled down all these proteins (Figure 4B). The interaction was highly specific compared to the two controls. These data suggest that the PSMA3 fragment active in binding p21 also binds a subset of IDPs. 

To obtain a more systemic view of the PSMA3 trapper domain, we used cellular extracts of over 1400 proteins in pull-down experiments. We identified 183 proteins that were retained in the GST–PSMA3 187–255 column and none in the control PSMA5 188-241. We termed the PSMA3 trapper binding proteins PSMA3–TBP. Previously, we had identified a group of highly disordered proteins that is readily degraded by the 20S proteasome in vitro and designated it as 20S-IDPome [18]. Interestingly, PSMA3–TBPs are significantly intrinsically disordered like the 20S–IDPome proteins (Figure 4C). Furthermore, out of 183 PSMA3–TBPs, 70 overlapped with the 20S–IDPome (Figure 4D). Random intersection between the groups was estimated based on a Z-test with the null distribution calculated by 10,000 simulations. Based on this calculation, the number of proteins shared by these two groups is highly significant over the random distribution (Figure 4E). These data suggest that a significant fraction of 20S–IDPome is trapped by PSMA3.

#### 3.2.3. Many of the PSMA3–TBPs Are Proteasome Substrates

The finding that some of the PSMA3–TBPs are 20S substrates may suggest that the trapper domain of the PSMA3 mediates degradation by the 20S proteasome in vitro. To substantiate this possibility, we conducted 20S degradation in vitro of cellular extracts in the presence and absence of the recombinant trapper domain (Figure 4F). We found that the recombinant GST–PSMA3 trapper markedly compromised 20S proteasomal degradation. The PSMA3 trapper did not inhibit proteasome catalytic activity, ruling out the possibility of the recombinant polypeptides inhibiting the 20S catalytic activity (Appendix A). The decoy effect was specific and was not repeated by the control-recombinant GST. These results suggest that in vitro, the 20S proteasomal degradation of many IDPs is regulated by the PSMA3 trapper (Figure 4G). 

#### 3.2.4. PSMA3–TBPs Sharing the 20S–IDPome Hallmarks Are 20S Substrates In Vitro

We previously reported that proteins of the 20S–IDPome are not only highly disordered, but also display a unique signature [18] in that they are significantly highly enriched for RNA binding proteins (RBPs), proteins with low complexity region (LCR) and proteins with prion-like domain (PrLD) [45]. Interestingly, the PSMA3–TBPs also have the 20S–IDPome signature (Figure 5A–E). 

The 20S–IDPome is enriched with phase-separation proteins [18]. Phase-separation proteins are characterized by pi-orbital-containing residues and are involved in pi–pi interactions [46,47]. Utilizing PScore, the phase separation predictive algorithm based on pi interaction frequency, we found significant enrichment for phase-separation proteins in both 20S–IDPome and PSMA3–TBPs (Figure 5F).

Next, we asked why some of the PSMA3–TBPs are not 20S substrates. To this end, we compared the PSMA3–TBPs that were identified as 20S substrates (70 proteins) to the rest (113 proteins) (Figure 5G–L). Consistent with our model, the PSMA3–TBP group of proteins that are not 20S substrates only poorly displayed the 20S–IDPome signature (Figure 5G–L). These data suggest that PSMA3–TBPs with the hallmark of the 20S–IDPome are the preferred 20S substrates.

#### 3.2.5. PSMA3–TBPs Sharing the 20S–IDPome Hallmarks Are Proteasome Substrates In Vivo

Since the 20S–IDPome group was identified based on in vitro degradation studies, the physiological relevance of the findings has to be validated. Therefore, we performed a similar study using a published dataset of cellular nascent substrates that were identified by virtue of their proteasomal association and degradation in cells [48]. Remarkably, many of the PSMA3–TBPs are cellular nascent proteasome substrates (Figure 6A), far more than would be expected by random distribution (Figure 6B). Furthermore, this shared group of proteins displays the 20S–IDPome signature (Figure 5C–G). Here again, the non-shared group lacks this signature. These data suggest that PSMA3–TBPs bearing the 20S–IDPome signature are more prone to proteasomal degradation in vivo. 

## 4. Discussion

Understanding the mechanisms of substrate recognition by the proteasome is critical for appreciating the importance of proteostasis. The mechanism controlled by substrate ubiquitination that mediates recognition by the 19S regulatory particle of the 26S proteasome is well known [10]. However, our knowledge of how the substrates are recognized by the 20S proteasomes, in the ubiquitin-independent degradation process, is very limited. Ornithine decarboxylase proteasomal degradation is ubiquitin-independent and mediated by antizyme [49], but this is not the case with other substrates. Here, we investigated the possibility that certain subunits of the 20S proteasome directly trap the relevant substrates for degradation. We provided evidence that the PSMA3 C-terminal region plays the role of trapper of a group of IDPs. We demonstrated that the trapper interacts in vivo with the IDP p21 and pulls down c-Fos and p53, and that trapper-less cells poorly degrade these proteins. We further showed that the recombinant trapper inhibits the degradation of proteins by the 20S proteasome, possibly by competing with the substrates to bind the trapper in its natural context. Interestingly, the trapper binds a large number of proteins sharing the hallmarks of the previously identified 20S substrates [18]. These findings are consistent with a model whereby the PSMA3 trapper recognizes the substrate for degradation.

PSMA3, also termed C8, has been reported to bind several proteins. These include p21 [43,50], Rb-MDM2 complex [51], Id-1, a member of the HLH protein family [52] and Epstein–Barr virus EBNA3 proteins [53]. Interestingly, in some instances, the PSMA3 interaction is facilitated by adaptor/regulator proteins. For example, 14-3-3t regulates p21 degradation [50], MDM2 regulates Rb degradation [51] and Id-1 promotes the 20S proteasomal degradation of HBx, the hepatitis B virus regulatory protein [52]. 

How the PSMA3 trapper recognizes a large number of IDPs is not known. The trapper C-terminal tail is highly acidic; however, the PSMA5 also has an acidic tail but is inactive in IDP binding. This is not because the PSMA5 is less accessible to the surface, as even the PSMA5 C-terminus in isolation did not bind IDPs. Although a charged tail might be important, it seems to not be sufficient. Additional structural analysis is required to resolve this relevant question.

Not all the trapper-binding proteins are optimal proteasome substrates in the process of ubiquitin-independent degradation. Interestingly, the trapper-binding proteins that are either degraded by the 20S proteasome in vitro or identified as proteasome nascent substrates in the cells, share the hallmarks of the 20S–IDPome; namely, they are IDP, RBP, GR/PR di-peptide repeats interactor proteins [45], containing a low complexity region (LCR) or a prion-like domain (PrLD). Trapper-binding proteins that were not well degraded by the proteasomes only poorly display this signature. Remarkably, components of liquid droplets share the same signature [18]. Furthermore, we found a significant enrichment for phase separation proteins in both 20S–IDPome and PSMA3–TBPs (Figure 5F). An interesting possibility is that the trapper preferentially recognizes liquid droplets for degradation. 

A key question is how the IDP trapper/degradation is regulated to permit substrate discrimination. The trapper region is post-translationally modified, particularly at the 250 residues (https://www.phosphosite.org/homeAction.action. Accessed date 28 January 2020). Additionally, it has been clearly demonstrated that IDPs undergo rather extensive post-translational modifications [54,55,56]. These modifications, largely S/T phosphorylation, might regulate trapper binding and degradation. Additionally, previously, we reported on nanny proteins that interact with IDPs to help escape their proteasomal degradation [57]. Nanny proteins therefore might function to help the substrate in escaping the trapper. According to these models, trapper interaction takes place by default unless inhibited. This model is substantially different from the ubiquitination model wherein the substrate is stable unless marked by ubiquitin. 

## Figures and Tables

**Figure 1 cells-11-03231-f001:**
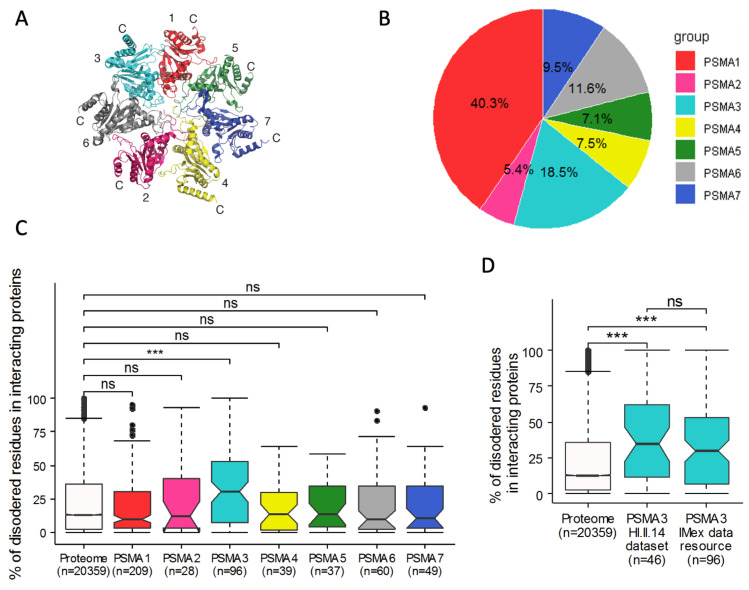
PSMA3 preferentially interacts with IDPs. (**A**) Crystal structure of PSMA ring [38]. PSMA subunits are identified by numbers. The N termini of the PSMA subunits protrude into the center of the ring, forming a gate restricting access into the 20S proteasome. (**B**) Pie chart presenting identified protein interactions of each PSMA subunit as a percentage of all identified protein interactions with PSMA subunits. We used the IMEx data resource to assemble an interaction list for the subunits. (**C**,**D**) Boxplot presenting the fraction of disordered residues found in the interacting proteins’ sequences. Non-overlapping notches provide a 95% confidence level that medians differ. Disordered residues were predicted with the IUPred algorithm. (**C**) Comparison of the level of disorder in proteins interacting with different PSMA subunits using IMEx data resource. (**D**) Distribution of PSMA3-interacting proteins from HI.II.14 dataset and IMEx data resource. The PSMA subunits are color-coded. ns (non-significant). *** *p* ≤ 0.001.

**Figure 2 cells-11-03231-f002:**
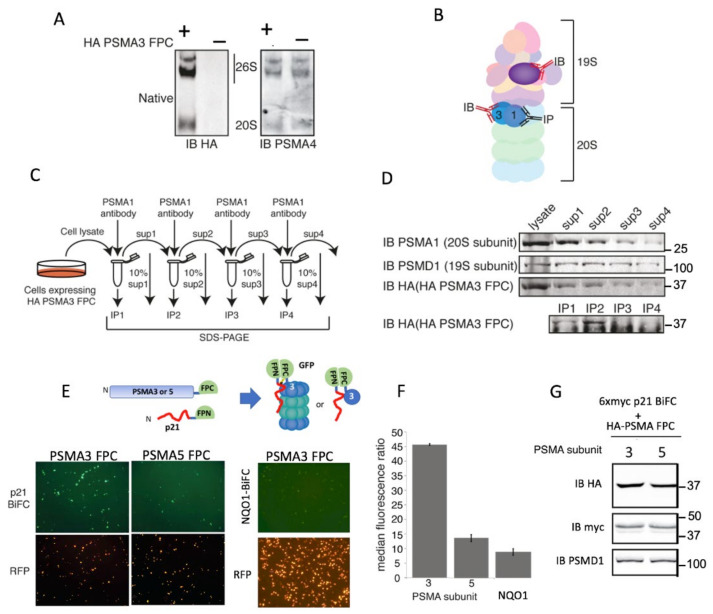
PSMA3 interacts with p21 in the cells. (**A**) U2OS cells stably expressing PSMA3–FPC. Cell lysates were enriched with proteasomes by ultracentrifugation and loaded on native gel. The membrane was probed with antibody against either HA-tag or the endogenous subunit PSMA4. (**B**) Schematic description of the antibodies used against the different 26S proteasome subunits for the described co-immunoprecipitation experiments to demonstrate the possible incorporation of a chimeric PSMA3 subunit into proteasomes. The endogenous PSMA1 subunit was first immunoprecipitated and the level of the co-immunoprecipitated subunits was monitored using antibodies to detect the endogenous PSMD1, a subunit of the 19S proteasome, and anti-HA to detect the chimeric PSMA3. (**C**) The schematic description of the experimental strategy of serial consecutive immunoprecipitation steps. (**D**) The results obtained from each of the steps described in Panel C. HEK293 cells expressing HA–PSMA3–FPC were harvested 24 h post-transfection. Cells’ lysate was subjected to four subsequent immunoprecipitations of proteasomes via the endogenous PSMA1 subunit. Ten percent of cell lysate was kept for analysis after each immunoprecipitation. (**E**) Cells were transfected with either PSMA3–FPC or PSMA5–FPC together with p21 FPN (see scheme). We also transfected the cells with H2B–RFP, which provides RFP labeling of the transfected cells’ nuclei. Successful BiFC using fluorescent microscopy, 20× objective 48 h post-transfection. (**F**) Intensities of at least 10,000 cells for each PSMA–p21 and PSMA–NQO1 combination were recorded by flow cytometry. Standard deviation bars represent two biological replica. (**G**) Expression level of the proteins in the cells was examined.

**Figure 3 cells-11-03231-f003:**
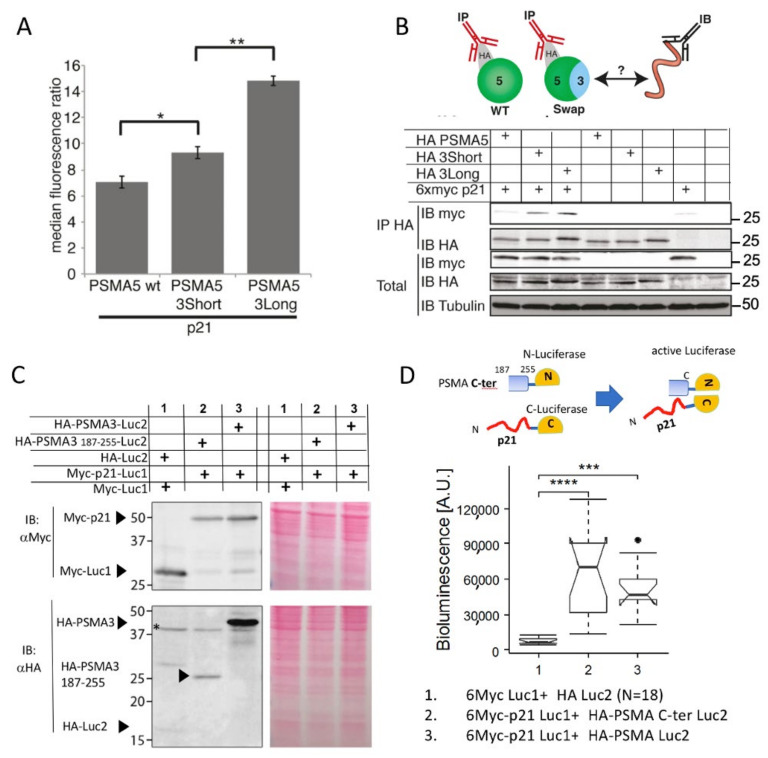
The PSMA3 C-terminus is sufficient to interact with p21. (**A**) Cells were transfected with the PSMA5 and three chimera (see Appendix A) together with p21-VFP and fluorescence intensities of at least 10,000 cells for each case were monitored and recorded by flow cytometry. PSMA5-3short is made of PSMA5 amino acids 1–187 with PSMA3 (187-229 aa) at its c-terminus. PSMA5-3long is as above but the PSMA3 fragment is longer (187–255 aa). Standard deviation bars represent three independent experiments. * *p* = 0.03, ** *p* = 0.001 using a two-tailed Student *t* test. (**B**) Schematic illustration of the experimental strategy with the antibodies used for immunoprecipitation (IP) or immunoblotting (IB), the latter to detect myc-tagged p21. HEK293 cells were transiently transfected as indicated with 6xmyc p21 and chimeric PSMA5 subunits. Cells were harvested 48 h post-transfection, lysed and subjected to IP with HA beads to immunoprecipitate chimeric PSMA5 subunits. Total lysate and IP samples were analyzed by SDS–PAGE and immunoblotting. (**C**) Representative SDS–PAGE and immunoblot analysis of the overexpressed proteins that were tested for interaction. Ponceau staining was used as a loading control. * non-specific band (**D**) The scheme of divided luciferase experiment is shown above the obtained data analyzed by Boxplot representing the overall bioluminescent signals corresponding to interaction between p21 and either PSMA3 or PSM3 187–255. The samples are numbered based on Panel C. n = 14–18. *** *p* ≤ 0.001, **** *p* ≤ 0.0001, using a two-tailed Student *t* test.

**Figure 4 cells-11-03231-f004:**
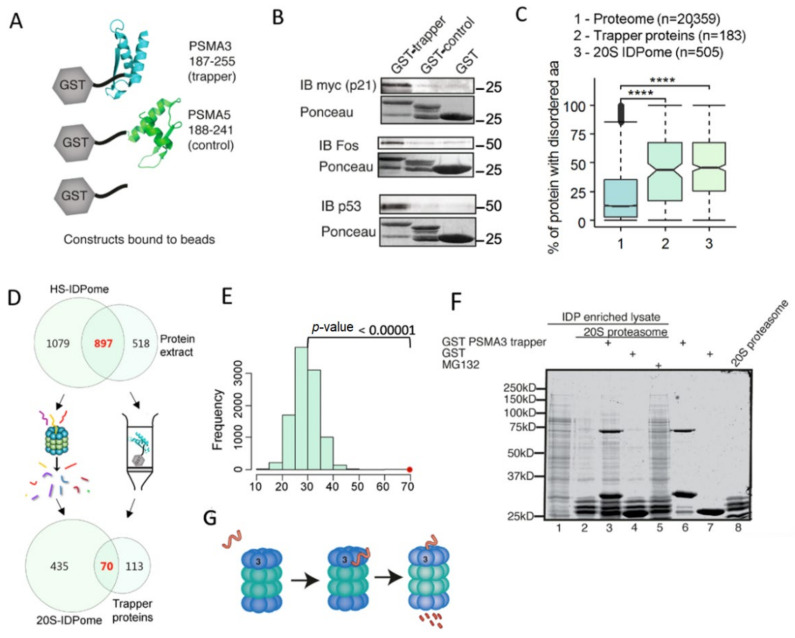
Isolated PSMA3 C-terminus interacts with many intrinsically disordered proteins. (**A**) Illustration of constructs used and experimental strategy. (**B**) Purified GST, GST PSMA3 trapper and GST PSMA5 C terminus (control) bound to glutathione agarose beads were incubated with HEK293 cell lysate overexpressing 6xmyc p21 or naive HEK293 cell lysate. GST constructs and the interacting proteins were eluted with 10 mM reduced glutathione. GST constructs were visualized with Ponceau and interacting proteins; myc-p21 and endogenous c-Fos and p53 were detected by immunoblot (IB). (**C**) The different GST-chimeric proteins described in A were incubated with cellular extract. The bound proteins were identified by MS. One hundred fifty-seven proteins were retained on the GST–PSMA3 trapper fragment, whereas only nine were retained on the GST–PSMA5 C terminus fragment. The former group was analyzed for IDP/IDR content. The Boxplot shows the IDP/IDR content of the proteome compared to the trapped proteins by the PSMA3 C-terminal fragment and to the 20S IDPome group. **** *p* ≤ 0.0001 (**D**) Venn diagram of the proteins retained on the column containing the GST–PSMA3 trapper fragment and the 20S IDPome. (**E**) The expected average number of shared proteins between the two groups in Panel D was evaluated by a Z-test with the null distribution calculated by 10,000 simulations. The expected intersection number is approximately 25 proteins, whereas the observed number is 70 (red dot, *p*-Value < 0.00001). (**F**) HEK293 IDP-enriched lysate was incubated for three hours at 37 °C with purified 20S proteasome, GST–PSMA3 trapper and GST as indicated. Protocol for 20S purification was previously described [18]. Proteins were visualized with InstantBlue stain. (**G**) A model describing the steps of IDP recognition by the trapper and degradation by the 20S.

**Figure 5 cells-11-03231-f005:**
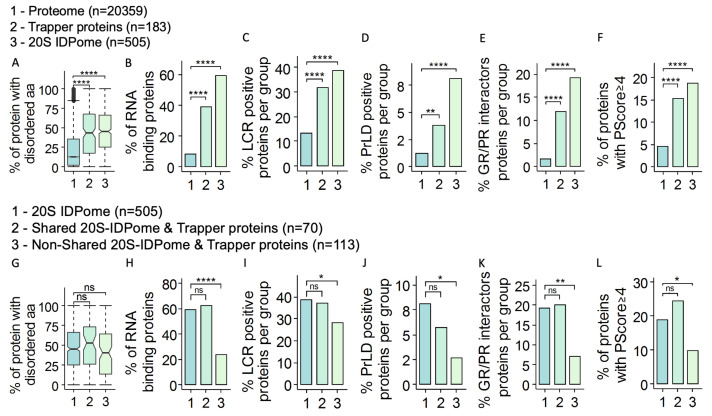
Many of the PSMA3–TBPs share the 20S proteasome substrate hallmarks (**A**–**F**) Three groups of proteins are compared and labeled -1 for overall proteome, 2 for PSMA3–TBPs and 3 for 20S–IDPome. These groups of proteins were compared for: (**A**) the average disorder degree (Boxplot); (**B**) percentage of RBPs in the group; (**C**) percentage of proteins positive for low complexity region (LCR); (**D**) percentage of proteins positive for prion-like domain (PrLD); (**E**) percentage of proteins interacting with GR/PR di-peptide repeats; (**F**) percentage of proteins with PScore equal to or above 4. (**G**–**L**) Under this set of panels, the 20S IDPome group of proteins (Group 1: 505 proteins) is compared to a fraction of the PSMA3–BPs that is shared by Group 1 (group 2: 70 proteins) and that is not shared by Group 1 (Group 3: 113 proteins). The comparison was conducted for: (**G**) the average disorder degree (Boxplot); (**H**) percentage of RBPs in the group; (**I**) percentage of proteins positive for low complexity region (LCR); (**J**) percentage of proteins positive for prion-like domain (PrLD); (**K**) percentage of proteins positive for GR/PR di-peptide repeats interactor proteins; and (**L**) percentage of proteins with PScore equal or above 4. NS (non-significant) *p* > 0.05, * *p* ≤ 0.05, ** *p* ≤ 0.01, **** *p* ≤ 0.0001, using a two-tailed Student *t* test.

**Figure 6 cells-11-03231-f006:**
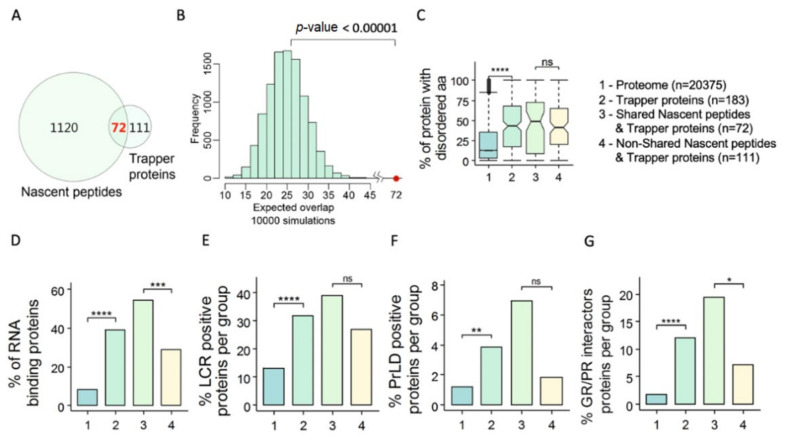
Many of the PSMA3 trapper-binding proteins (TBPs) are proteasomal associated in the cells (**A**) Venn diagram of the nascent proteasome substrates (nascent peptides, 1192 proteins) and the PSMA3–BPs (trapper). (**B**) The expected overlap between the two groups in Panel A was evaluated by a Z-test with the null distribution calculated by 10,000 simulations. The expected average overlapped group is 20 proteins, whereas the observed number is 72 (red dot) (*p*-value < 0.00001). (**C**–**G**) Under this set of panels, the proteome (Group 1) is compared to the following groups: PSMA3–TBPs (Group 2 n = 183); proteins that are found in Group 2 and also in the nascent substrate proteins (Group 3 n = 72); proteins that are of PSMA3–TPs group but not found in the nascent substrate proteins (Group 4 n = 111). The comparison was conducted for: (**C**) the average disorder degree (Boxplot); (**D**) percentage of RBPs in the group; (**E**) percentage of proteins positive for low complexity region (LCR); (**F**) percentage of proteins positive for prion-like domain (PrLD); and (**G**) percentage of proteins positive for GR/PR di-peptide repeats interactor proteins. NS (non-significant) *p* > 0.05, * *p* ≤ 0.05, ** *p* ≤ 0.01, *** *p* ≤ 0.001, **** *p* ≤ 0.0001, using a two-tailed Student *t* test.

## Data Availability

Not applicable.

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
