# Peer review of "The C-Terminus of the PSMA3 Proteasome Subunit Preferentially Traps Intrinsically Disordered Proteins for Degradation"

_cells, 2022, doi:10.3390/cells11203231_

Round 1

Reviewer 1 Report

The PSMA3 C-terminal region 19 is proposed by the authors of this MS to trap a subset of IDPs to facilitate their proteasomal degradation. This is a very interesting study, although many of the points need additional argument and investigation. Here are some of my concerns.

1- Please provide more details, including ID, company, and city, for all reagents used in this study, such as antibodies, chemicals, and so on.

2- To better evaluate the data in terms of figure resolution and blots, none of the figures in this MS are clear. Author could increase the size of images and text in figures to prepare for better, higher quality figures.

3- Why isn't there an increase in the protein level by endogenous PSMA4 in Figure 2A?

4- In blots, all proteins should be defined by their molecular weight.

5- Figure 2G is not clear, what is 3 and 5

6- Please change tissue culture to cell culture

7- (HEK293 cells were 68 transfected using the calcium phosphate method): What this refers to Please provide more information.

8- We assumed that an inherent component of the 20S complex could potentially act as 152: Please provide an explanation using any previous literature.

9- lines 176-184: I should see these outcomes in figures.

10- Line 200: see scheme: Please include it in the figure instead of saying "see scheme."

11- It is difficult to recognize the findings in Figure 5.

12- Discussion is inadequately presented, and the purpose of the work should be fully obvious.

13- The replicate of all blots data should be shown in supplementary figures 

Reviewer 2 Report

The manuscript of Assaf Biran et al. presents very interesting results, shedding light on how proteins with disordered structure or with disordered regions can be targeted for degradation by the proteasome.

The work is worth publishing after minor revision.

Reviewer 3 Report

The authors present a very nice discovery on how disordered proteins' ubiquitin-independent proteasomal degradation could be facilitated by the C-terminal disodered tail of one of the proteasome subunits, PSMA3. I think the presented work is very interesting, important and of high quality.

I only found very minor things to correct:

1) Figure 1 panel C Y axis title should be changed as "% of disodered residues in interacting proteins".

2) Fig 1 legend "predicated" should be predicted

3) Fig 1C legend: "Distribution of PSMA subunits interacting proteins from IMEx data resource" should be rephrased such as "Comparison of the level of disorder in proteins interacting with different PSMA subunits."

4) line 225:

"truncation of the 187-255 C-terminus region" should be rephrased as

" truncation of the C-terminal region (residues 187-255)".
5) line 294 "C-terminus fragment" should be "C-terminal fragment"

6) line 320:

"and GR/PR di-peptide repeats interactor proteins" should be rephrased as  "and proteins interacting with GR/PR di-peptide repeats"

7) lines 328-329 "percent of proteins positive for GR/PR di-peptide repeats interactor proteins" should be rephrased as "percent of proteins interacting with GR/PR di-peptide repeats"

8) line 335 present of proteins --> percent of proteins

9) line 381

PSMA3 C-terminus region --> PSMA3 C-terminal region

10) line 383 "degrades" should be "degrade" because cells was in plural

11) line 396 C-terminus tail --> C-terminal tail

12) line 406

"and a prion-like domain (PrLD)" should be "OR a prion-like domain (PrLD)"

13) line 418 "to escape their proteasomal degradation" should be " to help escape their ..."
